# A NEURAL NETWORK FRAMEWORK FOR LEARNING GREEN'S FUNCTION

## ABSTRACT

Green's function plays a significant role in both theoretical analysis and numerical computing of partial differential equations (PDEs). However, in most cases, Green's function is difficult to compute. The troubles arise in the following three folds. Firstly, compared with the original PDE, the dimension of Green's function is doubled, making it impossible to be handled by traditional mesh-based methods. Secondly, Green's function usually contains singularities which increase the difficulty to get a good approximation. Lastly, the computational domain may be very complex or even unbounded. To override these problems, we leverage the fundamental solution, boundary integral method and neural networks to develop a new method for computing Green's function with high accuracy in this paper. We focus on Green's function of Poisson and Helmholtz equations in bounded domains, unbounded domains and domains with interfaces. Extensive experiments illustrate the efficiency and the accuracy of our method for solving Green's function. In addition, we also use the Green's function calculated by our method to solve a class of PDE, and also obtain high-precision solutions, which shows the good generalization ability of our method on solving PDEs.

## 1 INTRODUCTION

Green's function plays an important role in the theoretical research and engineering application of many important partial differential equations (PDEs), such as the Poisson equation, Helmholtz equation, Maxwell equation, and so on. For one thing, Green's function can help solve PDE problems. When Green's function is known, the solutions of a class of PDE problems can be written explicitly in an integral form where Green's function serves as the integral kernel. Therefore, Green's functions can be regarded as the solution operators of PDEs mapping PDE data to the solution. For another, this integral representation makes Green's function a powerful tool to help study the analytical properties of the PDEs in theoretical research. Therefore, numerical methods to compute the Green's function have drawn great attention over the years.

However, there are mainly three problems in computing Green's functions. Firstly, solving Green's functions is a high-dimensional problem. Compared with the original PDE, its dimension is doubled, which limits the application of traditional methods to solve Green's function directly. At the same time, the computation cost is much higher than that of solving the original equation. Secondly, the derivative of Green's function is a Dirac delta function. Therefore, singularity exists in Green's functions, making it difficult to obtain a high-precision solution directly. Thirdly, the PDE may be defined on a very complex domain. In addition to the complex boundary, the domain may also be unbounded or has an interface, which also further increases the difficulties in the computation of Green's function.

Fortunately, with the rapid development of neural network and deep learning in recent years, machine learning has made important progress in many fields such as image recognition, natural language processing and so on (Goodfellow et al., 2016). At the same time, due to the universal approximation ability of neural networks and its great generalization ability for high-dimensional functions, there are numerous works attempting to solve problems in scientific computation with neural networks. Among them, many works focusing on solving parametric equations and learning solution operators are closely related to Green's function, which can also be seen as a solution operator, such as FNO (Li et al., 2020), DeepONet (Lu et al., 2019), Deep Green (Gin et al., 2020). In

addition, there are also works, like MOD-net (Zhang et al., 2021), using neural network to compute Green's function directly.

However, due to the complexity of Green's function and the learning of solution operators, there are three main issues in most of the existing work to be solved. First, most of these works of learning PDE solution operators or Green's function are based on supervised learning, which requires a large amount of solution data as the supervisory signal. The data needed is often obtained by solving a large number of PDEs through traditional methods or via actual measurements. However, in practice, the accurate solution data set may be difficult to obtain, and the computation cost for the traditional method to solve the equation is relatively high. Second, the generalization ability of these methods is not good enough. Because these works require exact solutions or parameters and the source terms of PDEs, the learned solution operator is only an approximation of Green's function, and the solution on the set outside the area covered by the training set will become worse. Last, for some problems such as electromagnetic wave propagation, solving PDEs in an unbounded domain is critical. However, existing methods also use a neural network to approximate the solution operators or Green's function directly, which severely suffer from the difficulty of sampling in unbounded domains.

To address these issues and overcome the three difficulties in computing Green's function mentioned above, in this paper, we design a novel neural network based formulation to compute Green's function directly. Firstly, we use the fundamental solution to remove the singularity of the Green's function, such that the equation for Green's function is reformulated into a general smooth high-dimensional equation. Then, neural network based methods are designed to solve this high-dimensional problem which is difficult to solve by traditional methods. In particular, we introduce two neural network formulation for this problem: derivative based method and boundary integral equation based (BIE-based) method. The idea of derivative based method is similar to PINN(Raissi et al., 2019), DGM (Sirignano & Spiliopoulos, 2018) and some other articles (Berg & Nyström, 2018), which use the residuals of equations and boundary conditions as the loss function, and directly approximate the objective function with a neural network, and calculate each order derivative of the network by automatic differentiation. BIE-based is based on BINet (Lin et al., 2021), which is a method for solving parameterized PDEs. BIE-based method transform the PDEs in the whole domain into the boundary integral equation such that the PDE is automatically satisfied and no extra differentiation with respect to network input is needed. It not only performs better than the derivative based method, but also can compute the Green's function in an unbounded domain, which is also the solution operator for PDEs in unbounded domain.

In conclusion, the main advantages of our work are summarized as follows: First, we propose a formulation to compute Green's function directly, which is a difficult problem for traditional methods to deal with. In addition, we also apply the Green's function computed by our formulation as the solution operator to solve a class of PDEs with high accuracy. Second, comparing with other methods of learning solution operators of PDEs, exact solutions are not required as the data set in our formulation. It not only reduces the complexity of preparation, but also improves the generalization ability of the calculated Green's function as the solution operator. Third, although the Green's function is very complex, we make full use of the properties of the Green's function itself, so we do not need to design a complex hidden layer network for approximation. Last, our formulation can not only compute the Green's function in any bounded domain, but also of the interface problems. In particular, by utilizing the boundary integral equation, the BIE-based formulation can also solve the Green's function in unbounded domains.

## 2 RELATED WORK

The study of Green's function has always been a very important issue, and there has been a lot of work. These works such as Duffy (2015), Greenberg (2015) introduce a lot of applications of Green's function. Because of the importance of the Green's function, computing Green's function is also a critical problem. In the notes (Hancock, 2006), authors gave the analytical expression of Green's function of the Poisson equation of some simple cases, and the paper (Kukla et al., 2012) discussed Green's function of the Helmholtz equation in or out of the unit disc. However, there is no good method to calculate the Green's function in the general domain.

An important application of Green's function is that it can solve a class of PDE, so it can be seen as a solution operator. In recent years, with the development of the deep learning, there has also been a lot of works on learning solution operators of PDEs (Hsieh et al., 2019; Li et al., 2020; Lu et al., 2019; Gin et al., 2020). Because of the strong representation ability of neural networks, many works did not want to get only the mapping of spatial coordinates to solutions. They use the neural networks to find the mapping from parameter space to the solution space. The idea of the works (Li et al., 2020; Gin et al., 2020) is closely related to Green's function. In addition, there is also workZhang et al. (2021) using the idea of the Green's function to calculate the solution operator. However, in the training process, these methods are difficult to avoid the dependence on the exact solution data and the information of the PDEs such as source term and boundary conditions.

## 3 FORMULATION

### 3.1 PROBLEM DESCRIPTION

In this paper, for simplicity, we focus on the Dirichlet Green's function, but the other type Green's function can be easily handled in GreenNet with a small modification. We consider both problems defined on a single domain, which can be further divided into interior problem and exterior problem, and the interface problem defined on two domains with different PDE parameters separated by a interface, as will be elaborated below.

#### 3.1.1 GREEN'S FUNCTION ON A SINGLE DOMAIN

Assuming $\Omega \subset \mathbb{R}^d$ is a bounded domain, and $\Omega^c = \mathbb{R}^d \backslash \Omega$. The interior problem and the exterior problem are formulated as

- Interior problem,

$$\begin{cases} \mathcal{L}u(x) = f(x) \text{ in } \Omega, \\ u(x) = g(x) \text{ on } \partial\Omega, \end{cases} \tag{1}$$

- Exterior problem,

$$\begin{cases} \mathcal{L}u(x) = f(x) \text{ in } \Omega^c, \\ u(x) = g(x) \text{ on } \partial\Omega + \text{ some boundary conditions at infinity}, \end{cases} \tag{2}$$

where $\mathcal{L}$ is a differential operator. For brevity, we represent the equations in (1) and (2) as

$$\mathcal{L}u(x) = f(x) \text{ in } \Omega^*, \tag{3}$$

where $\Omega^* = \Omega$ or $\Omega^c$ for the interior and exterior problems respectively.

In this paper, we focus on the Poisson equations and Helmholtz equations, i.e. $\mathcal{L} = -\Delta$ or $-\Delta - k^2$, where $k$ is the wave number. Helmholtz equations are the expansion of wave equation in frequency domain. It is used to describe wave propagation and is widely used in electromagnetics and acoustics. Because it involves wave propagation, Helmholtz equation often appears in the problem in unbounded domain, and because of the instability of Helmholtz equation, its numerical solution has always been an important problem.

For the equation (3), we can use the corresponding Green's function $G(x)$ to find the analytical solution $u(x) = \int_{\Omega^*} G(x,y)f(y)dy + \int_{\partial\Omega} \frac{\partial G(x,y)}{\partial n_y}g(y)ds_y$, where $G(x,y)$ is a $2d$-dimensional function satisfying

$$\begin{cases} \mathcal{L}_y G(x,y) = \delta(x-y), & \forall x, y \in \Omega^*, \\ \quad\quad\; G(x,y) = 0, & \forall x \in \Omega^*,\ y \in \partial\Omega. \end{cases} \tag{4}$$

#### 3.1.2 GREEN'S FUNCTION OF THE INTERFACE PROBLEM

Suppose $\Omega \subset \mathbb{R}^d$, which can either be a bounded domain or just $\mathbb{R}^d$. An interface $\Gamma \subset \mathbb{R}^{d-1}$ divides $\Omega$ into two regions, i.e., inside ($\Omega^-$) and outside ($\Omega^+$) of the interface, such that $\Omega = \Omega^- \cup \Omega^+ \cup \Gamma$. The interface problem is then formulated as:

$$\begin{cases} \mathcal{L}u = f, & \text{in } \Omega, \\ [u] = g_1, \quad \left[\dfrac{1}{\mu}\dfrac{\partial u}{\partial n}\right] = g_2, & \text{on } \Gamma. \end{cases} \tag{5}$$

where $\mathcal{L}$ is an operator with different parameters inside and outside the interface $\Gamma$, $n$ is the outward normal vector on the interface $\Gamma$. The bracket $[\cdot]$ denotes the jump discontinuity of the quantity approaching from $\Omega^+$ minus the one from $\Omega^-$. Similar to the single domain case, we focus on

- Poisson equations: $\mathcal{L}u = -\nabla \cdot (\frac{1}{\mu}\nabla u)$, where $\mu$ is a piecewise constant parameter such that $\mu = \mu_1$ in $\Omega^-$ and $\mu = \mu_2$ in $\Omega^+$;
- Helmholtz equations: $\mathcal{L}u = -\nabla \cdot (\frac{1}{\mu}\nabla u) - \varepsilon k^2 u$, where $\mu$ and $\epsilon$ are also piecewise constant parameters such that $\mu = \mu_1$, $\varepsilon = \varepsilon_1$ in $\Omega^-$ and $\mu = \mu_2$, $\varepsilon = \varepsilon_2$ in $\Omega^+$.

Moreover, some boundary condition on $\partial\Omega$ for bounded $\Omega$ (or at infinity for $\Omega = \mathbb{R}^d$) should be considered together to make the interface problem well-posed, which will be specified in detail in the experiments.

Utilizing the corresponding Green's function, the solution to the interface problem can be give by

$$u(x) = \int_\Omega G(x,y)f(y)dy + \int_\Gamma \left(\frac{1}{\mu}\frac{\partial G(x,y)}{\partial n_y}g_1(y) - G(x,y)g_2(y)\right)ds_y \tag{6}$$
$$+ \text{ item corresponding to the boundary condition on } \partial\Omega,$$

and $G(x,y)$ satisfies

$$\begin{cases} \mathcal{L}_y G(x,y) = \delta(x,y), & \forall\, x,y \in \Omega \\ [G(x,y)] = \left[\dfrac{1}{\mu}\dfrac{\partial G(x,y)}{\partial n_y}\right] = 0, & \forall\, x \in \Omega,\ y \in \Gamma, \end{cases} \tag{7}$$

along with some boundary conditions for $y \in \partial\Omega$, which will be specified in the experiments.

## 3.2 REMOVE THE SINGULARITY

As mentioned above, the Green's function is a singular function. However, we can change the equation (3) into a smooth equation by using the fundamental solutions $G_0$ of the original PDEs satisfiying $\mathcal{L}_y G_0(x,y) = \delta(x-y)$ for all $x,y \in \mathbb{R}^d$, which also can be seen as the Green's function of the whole space $\mathbb{R}^d$. For many important PDEs, fundamental solutions can be written explicitly. For the Poisson equation $-\Delta u(x) = f(x)$ in $\mathbb{R}^2$, the fundamental solution is $G_0(x,y) = -\frac{1}{2\pi}\ln|x-y|$, while the fundamental solution for the Helmholtz equation $-\Delta u(x) - k^2 u(x) = f(x)$ in $\mathbb{R}^2$ is $G_0(x,y) = \frac{i}{4}H_0^1(k|x-y|)$ where $H_0^1$ is the Hankel function.

For the Green's function on a single domain, we set $H(x,y) = G(x,y) - G_0(x,y)$, then $H$ is a smooth function and satisfies the following equation

$$\begin{cases} \mathcal{L}_y H(x,y) = 0, & \forall\, x,y \in \Omega^*, \\ H(x,y) = -G_0(x,y), & \forall\, x \in \Omega^*,\ y \in \partial\Omega. \end{cases} \tag{8}$$

For the Green's function of the interface problem, the fundamental solution of the Poisson equation is $G_0(x,y) = -\frac{\mu(y)}{2\pi}\ln|x-y|$, while the fundamental solution of the Helmholtz equation is $G_0(x,y) = \frac{i}{4}\mu(y)H_0^1(k\sqrt{\varepsilon(y)\mu(y)}|x-y|)$. Set $H(x,y) = G(x,y) - G_0(x,y)$, then $H$ satisfies the following equation

$$\begin{cases} \mathcal{L}_y H(x,y) = 0, & \forall\, x,y \in \Omega, \\ [H(x,y)] = -[G_0(x,y)], \quad \left[\dfrac{1}{\mu}\dfrac{\partial H(x,y)}{\partial n_y}\right] = -\left[\dfrac{1}{\mu}\dfrac{\partial G_0(x,y)}{\partial n_y}\right], & \forall\, x \in \Omega,\ y \in \Gamma, \end{cases} \tag{9}$$

along with some boundary conditions for $y \in \partial\Omega$. After the singularity of the original equation is successfully eliminated, we will introduce how to use neural network to solve this problem by combining boundary integral method and neural network.

### 3.3 NEURAL NETWORK APPROXIMATION

After removing the singularity, we transform the problem (4) into problem (8) and (9). Although the singularities have been removed, problem (8) and (9) are still of high dimension such that traditional methods are difficult to handle. Therefore, based on the Boundary Integral Network (Lin et al., 2021) and the Physics Informed Neutral Network (Raissi et al., 2019), we introduce two neural network-based methods. For simplicity, we take problem (8) as an example. The specific method for the interface problem can be obtained similarly.

#### 3.3.1 DERIVATIVE BASED-METHOD

For the interior problem in bounded domain, we can also use the PINN type method to solve the equaiton (8). We use a neural network $\tilde{H}(x, y; \theta)$ to approximate the function $H(x, y)$. Then we can calculate the derivative $\mathcal{L}_y \tilde{H}(x, y; \theta)$ by automatic differentiation. The loss function can be designed as the following form

$$L = \sum_{i=1}^{N} |\mathcal{L}_y \tilde{H}(x_i, y_i; \theta)|^2 + \sum_{j=1}^{M} |\tilde{H}(x_j, y_j; \theta) + G_0(x_j, y_j)|^2 \tag{10}$$

#### 3.3.2 BOUNDARY INTEGRAL EQUATION-BASED METHOD

However, the derivative based method is not stable, and the accuracy is poor. More importantly, it can not deal with the problems in unbounded domains. In the literature (Lin et al., 2021), authors introduced the BINet method, a boundary integral equation (BIE) based method to solve the parametric equation in the form of equation (8). In particular, we choose the double layer potential operator expression in our formulation. The neural network structure can be written as the kernel integral form $\mathcal{D}[h](x, y) := -\int_{\partial\Omega} \frac{\partial G_0(y,z)}{\partial n_z} h(x, z) ds_z$, where $G_0$ is the fundamental solution and the function $h(x, y) \in \Omega^* \times \partial\Omega$ is approximated by a hidden layer network, which can be selected as MLP or ResNet. The output of this network will satisfy the equation (8) automatically, and we only need to fit the boundary condition, which the loss function is based on. This is also the reason that this method can handle the exterior PDE problem in unbounded domain. The loss function is

$$L = \sum_{i=1}^{N} |\mathcal{D}[h](x_i, y_i) \mp \frac{1}{2} h(x_i, y_i) + G_0(x_i, y_i)|^2. \tag{11}$$

## 4 NUMERICAL RESULTS

In the implementation, we choose a ResNet structure neural network introduced in Deep Ritz method (Weinan & Yu, 2018). In the experiments, we choose the Adam optimizer to minimize the loss function, and we training the neural network on a single GPU of Tesla V100.

### 4.1 THE GREEN'S FUNCTION OF PDES IN BOUNDED DOMAINS

**The Poisson equation in the unit disc.** In this experiment, we consider the Green's function of the Poisson equation in the unit disc. From the notes(Hancock, 2006), we can know the Green's function of the unit disc of the Poisson equation has the explicit expression, $G(x, y) = \frac{1}{2\pi} \ln \frac{r}{r'\rho}$,where $y \in S^1$ and $y' = \frac{y}{|y|^2}$, and $r = |x - y|, \quad r' = |x - y'|, \rho = |y|$. Then we can compare the exact Green's function and the numerical Green's function calculated by GreenNet.

In the training process, for the BIE-based method, we choose a ResNet with 8 blocks with 100 neurons per layer and ReLU activate function. For every 500 epochs, 80 new $x$ are randomly generated. After after 50000 epochs of training with learning rate 0.00001, we randomly generate 100 $x$ and 800 $y$ and compute the $G(x_i, y_j)$ of these 80000 points $(x_i, y_j)$ to obtain an average relative $L_2$ error of the Green's function $G(x, y)$ of 4.66e-3. For the derivative-based method, we used the neural network of same size. For every 500 epochs, 40000 new $(x_i, y_i) \in \Omega \times \Omega$ 10000 new $(x_j, y_j) \in \Omega \times \partial\Omega$ are randomly generated for the PDE loss and boundary condition loss respectively. Similarly, after 50000 epochs, the relative $L_2$ error of the Green's function $G(x, y)$ is 5.54e-2.

Table 1: Relative $L^2$ error of the Green's function in section 4.1

| structure | Poisson equation | Helmholtz equation |
|---|---|---|
| BIE-based | 7.44e-3 | 8.56e-3 |
| Derivative-based | 5.54e-2 | 8.75e-1 |

(a) x=(0.8427,0.4386),                    (b) x=(0.2923,0.0674)

Figure 1: The difference between the exact Green's function $G(x,y)$ of the unit disc and Green's function calculated by derivative-based and BIE-based methods with fixed $x = (0.8427, 0.4386)$ and $x = (0.2923, 0.0674)$.

The figure 1 show the difference of $y$ with fixed $x$ between the exact Green's function $G(x,y)$ of the unit disc and Green's function calculated by derivative-based and BIE-based methods. We can see the relative $L^2$ error of BIE based method still small when $x$ is close to the boundary. However, the error of derivative based method increases significantly.

**The Helmholtz equation in the square.** In this experiment, we consider the Green's function of the following equation

$$-\Delta u(x) - 4u(x) = f(x) \text{ in } \Omega,$$
$$u(x) = 0 \text{ on } \partial\Omega, \tag{12}$$

where $\Omega = [-1,1]^2$. The network structure and training details of the two methods are consistent with the previous experiment. Because the Green's function of a Helmholtz equation is complex number, we add a output of the network to represent the imaginary part. In this example, for some fixed $x_i$, we solve the equation (8) about $y$ as the ground truth. It should be noted that the equation has to be solved again for each x, so the computation cost is very expensive. The table 1 also show the relative $L^2$ error between ground truth and the numerical solution of Green's function. We can find derivative-based method failed. This is because the oscillation of Helmholtz solution will increase the difficulty for derivative based method. The superiority of BIE based method over derivative based method is shown here.

Next, we show the ability on solving the Helmholtz equation in $\Omega$ by Green's function computed by BIE based method. We consider using the Green's function to solve the equations with different source terms $f$, where $f$ belongs to the following set $\{((c_1^2+c_2^2)\pi^2-4)f_1(c_1\pi x_1)f_2(c_2\pi x_2)|c_1,c_2 = 1,\cdots,5\}$, where $x = (x_1,x_2) \in \Omega$ and $f_1(a), f_2(a) \in \{\sin(a), \cos(a+\pi/2)\}$. It can be seen that $f$ has 100 combinations. The histgram of the figure 2 show the distribution of the relative $L^2$ error between the exact solutions and the solutions calculated by our Green's function with different $f$. The relative error of the solution of the equation is between 0.01 and 0.045, which shows the stability of solving the equation with Green's function.

## 4.2 THE GREEN'S FUNCTION OF PDEs IN UNBOUNDED DOMAINS

**The Helmholtz equation out of the bow-tie antenna.** In this experiment, we consider the Green's function of Helmholtz equation in unbounded domains. The neural network of this experiement is same as the previous and training process is also similar. The exterior PDE problems are common in scattering problems, which is closely related to Helmholtz equation. We consider the following

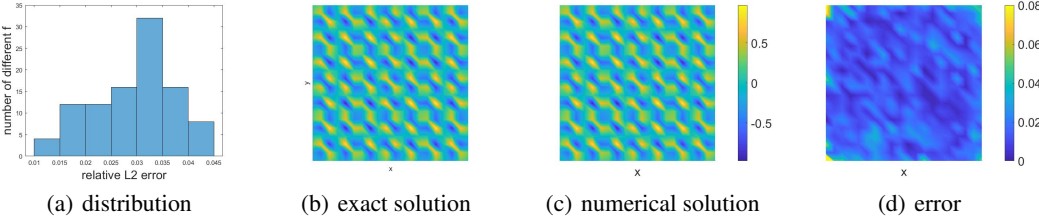

(a) distribution      (b) exact solution      (c) numerical solution      (d) error

Figure 2: (a) is the distribution of relative $L^2$ error of the solution of the PDE with 100 different source terms; (b) and (c) is the real part of the exact solution and the numerical solution with the source term $f = (72\pi^2 - 4)\cos(6\pi x_1 + \pi/2)\sin(6\pi x_2)$; (d) is the absolute error between exact solution and numercal solution.

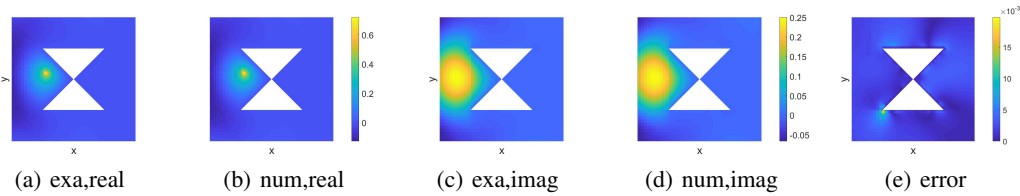

(a) exa,real      (b) num,real      (c) exa,imag      (d) num,imag      (e) error

Figure 3: For fixed x=(-0.4430,0.0938), (a) and (b) are the real part of the exact $G(x, y)$ and numerical $G(x, y)$ respectively. (c) and (d) are corresponding imaginary part. (e) is the absolute error between exact and numerical $G(x, y)$.

PDE problem,

$$-\Delta u(x) - 16u(x) = f(x) \text{ in } \Omega^c,$$

$$u(x) = 0 \text{ on } \partial\Omega, \quad \lim_{|x|\to\infty} (\frac{\partial}{\partial r} - 4i)u(x) = o(|x|^{-1/2}) \tag{13}$$

where the wave number in equation (13) is 4 and the limit of $|x|$ is the Sommerfeld condition. Let us consider a more practical scenario, a receiving antenna electromagnetic simulation problem. Assume we have a bow-tie antenna of 1 in length and 1 in width, which is a type of broad-bandwidth antenna. Its structure is 2 dimensional, implemented on a printed circuit board. Therefore, for simplicity, we consider simulating the field within the 2-D space. The shape of the bow-tie antenna can be seen in the figure 3.

We can also use our formulation to compute the Green's function in the domain outside the bow-tie antenna. Because the domain is unbounded, we can only used the BIE-based method. After training, the relative $L^2$ error of the Green's function is 4.98e-2. When the wave number increases and the domain of the PDE problem becomes more complex, a high-precision Green's function approximation is still obtained.

### 4.3 THE GREEN'S FUNCTION OF THE INTERFACE PROBLEM

**The Poisson equation in a square with flower-shaped interface** Let $\Omega$ be the square $\{(x, y) : |x| \leq 1, |y| \leq 1\}$, $\Gamma$ is parameterized by $(0.5\cos t - 0.1\cos 5t\cos t, 0.5\sin t - 0.1\cos 5t\sin t)$ with $t \in [0, 2\pi]$. 800 points are sampled on both $\Gamma$ and $\partial\Omega$ for boundary integral. Take $\mu_1 = 0.5, \mu_2 = 1$ in the Poisson equation. Dirichlet boundary condition is imposed on $\partial\Omega$ such that the Green's function to this problem satisfies (7) along with $G(x, y) = 0, \forall x \in \Omega, y \in \partial\Omega$.

In the training process, we choose a ResNet with 6 blocks with 100 neurons per layer. For every 500 epochs, 100 new $x$ is randomly generated. After after 100000 epochs of training with learning rate 0.00004, we fix 100 newly generated $x$ and obtain an average relative $L_2$ error of the Green's function $G(x, y)$ of 4.05%. Further more, for two fixed $x$, the exact Green's function (obtained by traditional boundary integral method), numerical solution obtained by neural network and absolute error is shown in Fig. 4. The results show that the proposed method can solve the Green's function for the interface problem accurately.

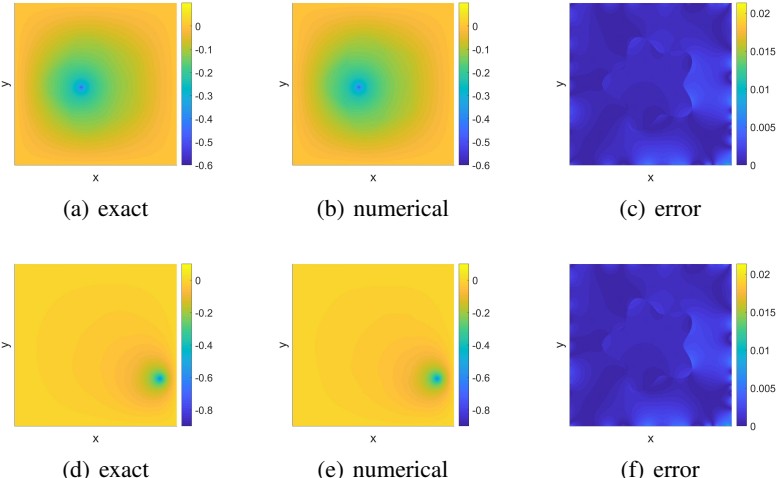

(a) exact      (b) numerical      (c) error

(d) exact      (e) numerical      (f) error

Figure 4: Figures (a)-(c) are for $x = (-0.1804, -0.0414) \in \Omega^-$ with relative $L^2$ error 1.94%, while figure (d)-(f) are for $x = (0.7899, -0.4157) \in \Omega^+$ with relative $L^2$ error 2.67%. First and second column: the exact solution and the numerical solution; Third column: absolute error between exact solution and numerical solution.

**The Helmholtz equation in $\mathbb{R}^2$ with square interface** Let $\Gamma$ be the square $\{(x, y) : |x| = 1 \text{ or } |y| = 1\}$, on which 800 points are sampled for the boundary integral. Take $\mu_1 = 2, \mu_2 = 1, \varepsilon_1 = 1, \varepsilon_2 = 4, k = 2$ in the Helmholtz equation. The Sommerfeld condition is required at infinity, i.e., $\lim_{|x|\to\infty}(\frac{\partial}{\partial r} - ik\sqrt{\varepsilon_2\mu_2})u(x) = o(|x|^{-1/2})$, implying that $\lim_{|y|\to\infty}(\frac{\partial}{\partial r} - ik\sqrt{\varepsilon_2\mu_2})G(x, y) = o(|x|^{-1/2})$, which is automatically satisfied if $H(x, y)$ is written in the boundary integral form.

The network and training hyperparameters are set the same as those in the Poisson equation example. The average relative $L_2$ error of the Green's function $G(x, y)$ is 4.63%. For two fixed $x$, the exact Green's function (obtained by traditional boundary integral method), numerical solution obtained by neural network and error is shown in Fig. 5, implying the effectiveness of the proposed method in computing Green's function.

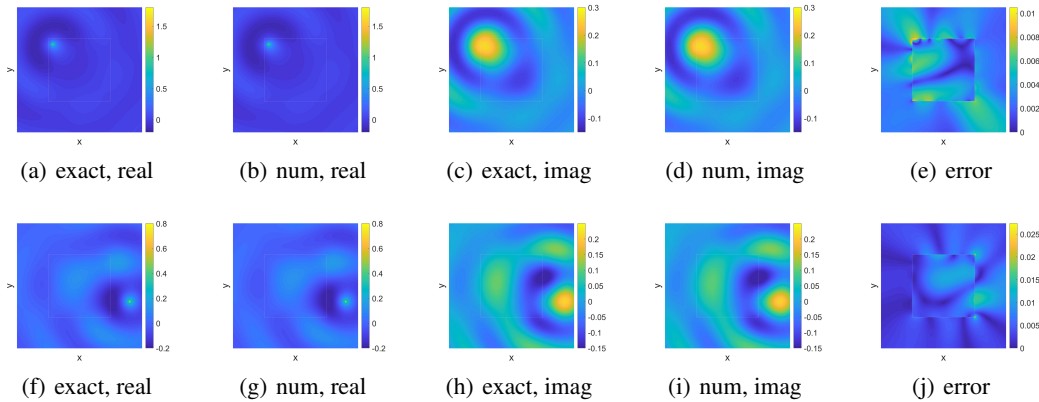

(a) exact, real    (b) num, real    (c) exact, imag    (d) num, imag    (e) error

(f) exact, real    (g) num, real    (h) exact, imag    (i) num, imag    (j) error

Figure 5: Figures (a)-(e) are for $x = (-0.8559, -0.8764) \in \Omega^-$ with relative $L^2$ error 3.98%, while figure (f)-(j) are for $x = (1.6069, -0.5065) \in \Omega^+$ with relative $L^2$ error 2.73%. First and second column: the real part of the exact solution and the numerical solution; Third and Forth column: imaginary part of the exact solution and the numerical solution; Fifth column: absolute error between exact solution and numerical solution.

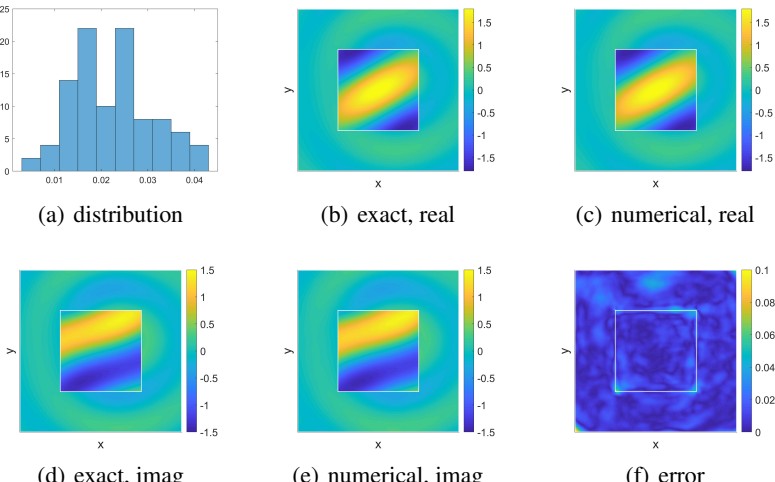

(a) distribution  (b) exact, real  (c) numerical, real

(d) exact, imag  (e) numerical, imag  (f) error

Figure 6: (a): histogram of the relative $L^2$ error of the 100 randomly generated equations. Figures (b)-(f) are the exact solution and numerical solution of one specific equation generated, whose parameters are given in the Appendix. (b) and (c): real part of the exact solution and the numerical solution; (d) and (e): imaginary part of the exact solution and the numerical solution; (f): absolute error between exact solution and numerical solution.

Furthermore, after the Green's function is learnt by the neural network, to show the generalization ability of the proposed method in solving PDEs, we consider the homogenuous case of problem (5), i.e., $f \equiv 0$, while the two jump conditions $g_1$ and $g_2$ are generated by the superposition of a class of parameterized function, which will be given in the Appendix. We randomly generate 100 sets of parameters, and solve the interface problem using the learnt Green's function. The histogram of the relative $L^2$ errors of the numerical solutions to the 100 equations are shown in Fig. 6 (a), while the solution and corresponding error for one set of parameters are given in Fig. 6. It can be seen that all the relative $L^2$ errors are below $4\%$. Therefore, not only can the proposed method accurately solve a class of PDEs accurately, but also this method has natural generalization ability over the PDE information.

## 5 CONCLUSION

In this paper, a novel neural network based method for learning Green's function is proposed. By utilizing the fundamental solution to remove the singularity in Green's function, the PDEs required for Green's function is reformulated into a smooth high-dimensional problem. Two neural network based methods are propsed to solve this high-dimensional problem. The first one, based on the idea of PINN, uses the neural network to directly approximate Green's function and take the residual of the differential equation and the boundary conditions as the loss. The second one is based on the recently proposed BINet, in which the solution is written in an boundary integral form such that the PDE is automatically satisfied and only boundary terms need to be fitted.

Extensive experiments are conducted and three conclusions can be drawn from the results. First, the proposed method can effectively learn the Green's function of Poisson and Helmholtz equations in bounded domains, unbounded domains and domains with interfaces with high accuracy. Second, BINet-based method outperforms the PINN-based method in the accuracy of Green's function and the capability to handle problems in unbounded domains. Third, the Green's function obtained can be utilized to solve a class of PDEs accurately, and shows good generalization ability over the PDE data, including the source term and boundary conditions.

Although the proposed method exhibits great performance in computing Green's function, the dependence on the fundamental solution hinders the application of this method in varying coefficient problems or equations without explicit fundamental solution.This will be further investigated in the future.

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

## APPENDIX

## A EXPERIMENTAL DETAILS

### A.1 HELMHOLTZ PROBLEM WITH INTERFACE

The exact solution to the interface problem is designed as

$$
u(x,y) = \begin{cases} \sum\limits_{i=1} c_1(i)e^{j(k_1(i)x+k_2(i)y)\sqrt{\varepsilon_1\mu_1}}, & (x,y) \in \Omega^-, \\ \sum\limits_{i=1} c_2(i)H_0^1(k\sqrt{\varepsilon_1\mu_1}\sqrt{(x-x_0(i))^2+(y-y_0(i))^2}), & (x,y) \in \Omega^+, \end{cases} \tag{14}
$$

where $\{c_1, c_2, k_1, k_2, x_0, y_0\}$ is a set of randomly generated parameters satisfying $k_1^2 + k_2^2 = k^2$ and $(x_0(i), y_0(i)) \in \Omega^-$, $\forall i$. Specifically, $c_1, c_2 \sim U[0, 1]$, $k_1 = k\cos\theta, k_2 = k\sin\theta$ with $\theta \sim U[0, 2\pi]$, $x_0, y_0 \sim U[-0.8, 0.8]$. $g_1$ and $g_2$ can be directly computed using the exact solution, and the solution to the PDE can be given by

$$u(x) = \int_\Gamma \left( \frac{1}{\mu} \frac{\partial G(x, y)}{\partial n_y} g_1(y) - G(x, y)g_2(y) \right) ds_y.$$

We take $I = 3$. The parameters of the exact solution corresponding to Fig. (6) (b)-(f) are given in the following table.

Table 2: Parameter setting for Fig. (6) (b)-(f)

| Index | $i = 1$ | $i = 2$ | $i = 3$ |
|---|---|---|---|
| $c_1$ | 0.5550 | 0.9934 | 0.2986 |
| $k_1$ | 1.9959 | 1.6667 | -0.9091 |
| $k_2$ | -0.1282 | -1.1056 | 1.7814 |
| $c_2$ | 0.3963 | 0.3051 | 0.3642 |
| $x_0$ | 0.2850 | 0.1625 | 0.2190 |
| $y_0$ | 0.5724 | -0.1154 | 0.6809 |

