# OpenReview forum: "A neural network framework for learning Green's function"
_ICLR.cc/2022/Conference — ICLR 2022 Submitted_

### Official Review · Reviewer_YLib · 2021-10-29

**Correctness:** 2
**Technical Novelty And Significance:** 1
**Empirical Novelty And Significance:** 1
**Recommendation:** 3
**Confidence:** 4

**Main Review:**

This is the wrong venue for this paper.  There is already a lot of expertise in solving PDEs.
Green's functions are a particular method for solving PDEs.  If this method is really better than state of the art methods for solving PDEs, the paper should be sent to a numerical analysis (PDE solving) journal, for a review by experts.
Strengths: the paper studies linear PDEs and compares a couple methods.
Weaknesses: the paper does not properly motivate the challenges solving PDEs, the relative advantages of using Green's functions, versus the many other methods out there.
Main weakness: how do we know that the methods solved here satisfy the basic requirements of any PDE solver: convergence.  Dnns are wildly unpredictable, so, in principle, anything can happen.

**Summary Of The Paper:**

This paper proposes to use neural networks to learn Green's functions.
Green's functions are a classical technique for solving linear PDEs.

**Summary Of The Review:**

This is a very narrow paper, which focuses on a particular method for solving linear PDEs.
However, there is very little motivation.  There are already decades worth of methods for solving PDEs. Using neural networks probably does worse than most off the shelf methods.  It would take a lot more evidence to be convincing that this method is better than the best standard PDE method for each problem.

---

### Official Review · Reviewer_Rxs8 · 2021-10-31

**Correctness:** 1
**Technical Novelty And Significance:** 2
**Empirical Novelty And Significance:** 2
**Recommendation:** 3
**Confidence:** 5

**Main Review:**

Green's functions of PDEs play important roles in numerous physical problems and their numerical computation is certainly an important problem and is worth to investigate. This paper made an interesting attempt for solving Green's functions using neural networks. However, from my perspective, the contribution of this paper is incremental and may not be sufficiently novel for ICLR. Here is a list of strength and weaknesses.

Strength

- The authors provide extensive numerical experiments for computing 2D Green functions of Poisson and Helmholtz problems.

Weaknesses

- The first weakness I have is that this work is that it does not discuss why neural networks is superior than the classical numerical methods such as  finite element method or finite difference method. I do not see any numerical comparisons between neural network methods and classical methods. In fact, since the authors mainly focus on Green's function on 2D, I do not see any advantage of using neural networks compared to these classical methods. My understanding is that neural networks can be very powerful for approximation high dimensional functions, but due to the lack of robustness in the training process, the neural network approximation can have low accuracy even in low dimensions, while in this setting FEM and FD are usually more robust and accurate. The authors need to justify the precise advantages of neural networks.

- The proposed method seems relying on that the singularity of the Green's function is known a priori so that one can separate that from approximation. However, what if such a priori information is not available? For example, can the authors comment on how the proposed method could be used to find the Green's function of Helmholtz equation in a non-homogeneous medium?

- Here the authors presented numerical results in 2D. I would recommend the authors provide at least one example for solving the Green's function of a high dimensional PDE (e.g. dim larger than 10). This way, one can possibly see the potential benefits of  neural networks.

Some typos and further comments

- In the second line below section 3.3.1,  "equaiton" should be "equation".
- Three lines below equation (12), " is complex number" should be " is complex". In the same line, "add a output of the network" should be "add an output of the network"
- Similar typos appear here and there in the paper. The authors should run a spell check before submitting the paper.

**Summary Of The Paper:**

This paper proposes a neural network based method for computing Green's functions of Poisson and Helmholtz equations defined on various domains. The idea is to first subtract the target Green's function with some background green's function which is assumed to be known and easy to evaluation. Then the next step is approximate by neural networks the difference function that satisfies the same PDE problem with nonhomogeneous boundary conditions. The authors compared two methods for solving the PDE problems, based on the PINN framework and the boundary integral equation framework.

**Summary Of The Review:**

To summarize, I regret that I am inclined to reject the submission since  the contribution of this paper is incremental and is not be sufficiently novel for ICLR.

---

### Official Review · Reviewer_mJPD · 2021-10-31

**Correctness:** 4
**Technical Novelty And Significance:** 1
**Empirical Novelty And Significance:** Not applicable
**Recommendation:** 1
**Confidence:** 5

**Main Review:**

I enjoyed reading the paper. Unfortunately, I do not think the approach is novel and the results are significant. The concept of using neural networks to approximate Green's function is not new. In addition, this method heavily depends on the existence of a fundamental solution hence is restricted in solving linear PDEs.

Furthermore, the error of the solution seems to be high as for both cases (Poisson and Helmholtz) the maximum error is about 4%. This is not satisfying especially considering the linearity of the problem. The author should compare their results to the classical boundary element method and I suspect that BEM can bring the error below 0.1% easily.



**Summary Of The Paper:**

The paper proposed a neural network based method for solving linear PDEs. The approach is based on approximating the solution using a combination of PINN and neural network based Green's function method. The architecture is then verified against the classical Poisson and Helmholtz equation.

**Summary Of The Review:**

In summary, I do not think the paper in its current form meets the standard of ICLR hence recommend a strong rejection.

---

### Official Review · Reviewer_dXB8 · 2021-11-02

**Correctness:** 3
**Technical Novelty And Significance:** 2
**Empirical Novelty And Significance:** 1
**Recommendation:** 3
**Confidence:** 3

**Main Review:**

Pros

- I find the formulation of the learning problem to be elegant, specifically: by way of using information about the PDE operator and inputing the fundamental solution and learning the residual ( H(.) )
- The proposed solution is unsupervised and does not require precomputed accurate solutions to the PDE’s

Cons

- I find the empirical evaluation of this paper to be significantly lacking. I see no comparisons with previous approaches like PINN, DeepGreen, DeepRitz etc. which are also deep learning solutions to solve similar PDEs. To me, even a starting comparison (error values and graphs similar to Figures 3-6) with standard solvers like Finite Difference/Finite Element based approaches would help in understanding the empirical contributions of this paper. As of now, even if I am inclined to accept that the proposed method works yielding low errors to challenging PDEs with different test-source terms, I have no idea if it works competitively to prior work.
- The necessity of inputing the fundamental solution is a big drawback.
- No results are shown for PDEs in higher dimensions, where the problem of estimating the greens functions is very significant, and hence the need for a “learned” solver like it is proposed here.

Questions:

- Is there any particular reason for choosing that specific source set for ‘test’ (Page 6)? What happens for source functions outside this set? does the method generalise? Discuss.
- What is the significance of the comparison between PINN and BINet? i.e. Why does the method outlined in this paper work well in conjunction with BINet and not PINN?
- It would be convincing to see a demonstration of learning a “high-dimensional” greens function using the method proposed in this paper.

**Summary Of The Paper:**

This paper explores a deep learning scheme to solve partial differential equations on regular domains (mostly 2D). This paper focuses on the method of Greens function for solving a class of linear PDE’s. More specifically: the Greens function corresponding to a PDE is a function that depends on (1.) the differential operator associated to the PDE (2.) the boundary and geometry of the domain on which the PDE is desired to be solved. Given this greens function, the solution to the PDE on this domain for *any* future input source function can be obtained quite simply by a convolution with the greens function (and taking care of boundary conditions). Therefore the authors motivate to “learn” this greens function using neural networks.

Furthermore, the authors propose an unsupervised formulation of this approach, by decomposing the greens function into a formulaic fundamental solution (G0) and a learnable residual ( H(.) ).  By way of simple algebraic computations, this function H(.) satisfies simpler constraints and is therefore proposed to be modelled with a deep network and optimised using two different schemes: (a) PINN - a loss function corresponding to the L2 error from constraints (b) BIE based method.

**Summary Of The Review:**

I find the idea proposed in this paper to be interesting, and the fact that the proposed method is unsupervised by carefully parametrizing a part of the PDE solution with neural networks is also a plus point. However, I am inclined to weigh heavily towards penalising this paper for a lack of an extensive empirical analysis, since I am not convinced on the efficacy of the proposed solution in comparison to existing approaches (learning or otherwise).

---

### Decision · Program_Chairs · 2022-01-20

**Decision:**

Reject

**Comment:**

All reviewers vote for rejecting this paper. The main points of criticism shared by the reviewers are missing novelty and missing/unclear significance of the contribution. There was no rebuttal, so this is a clear reject.